# Increased Serum Interleukin 10 Levels Are Associated with Increased Disease Activity and Increased Risk of Anti-SS-A/Ro Antibody Positivity in Patients with Systemic Lupus Erythematosus

**DOI:** 10.3390/biom13060974

**Published:** 2023-06-11

**Authors:** Agnieszka Winikajtis-Burzyńska, Marek Brzosko, Hanna Przepiera-Będzak

**Affiliations:** 1Individual Laboratory for Rheumatologic Diagnostics, Pomeranian Medical University in Szczecin, Unii Lubelskiej 1, 71-252 Szczecin, Poland; agawb@o2.pl; 2Department of Rheumatology, Internal Medicine, Geriatrics and Clinical Immunology, Pomeranian Medical University in Szczecin, Unii Lubelskiej 1, 71-252 Szczecin, Poland; marek.brzosko@pum.edu.pl

**Keywords:** systemic lupus erythematosus, IL-10, IL-6, anti-SS-A/Ro antibodies, disease activity, organ involvement

## Abstract

Interleukin 10 (IL-10) plays a role in inflammation and cell-type responses. The anti-SS-A/Ro antibody contributes to leucopenia, and cutaneous and neonatal lupus. Objectives: To evaluate the association between serum IL-10 levels and autoantibodies, disease activity and organ involvement in systemic lupus erythematosus (SLE) patients. Patients and methods: We studied 200 SLE patients and 50 controls. We analyzed organ involvement, disease activity, serum IL-10 and interleukin-6 (IL-6) levels, and antinuclear and antiphospholipid antibody profiles. Results: Serum IL-10 and IL-6 levels were higher in SLE patients than in controls (all *p* < 0.00001). Serum IL-10 levels were positively correlated with IL-6 (*p* < 0.00001), CRP (*p* < 0.00001), fibrinogen (*p* = 0.003), and ESR (*p* < 0.00001), and negatively correlated with hemoglobin (*p* = 0.0004) and lymphocytes (*p* = 0.01). Serum IL-6 levels were positively correlated with CRP (*p* < 0.00001), fibrinogen (*p* = 0.001), and ESR (*p* < 0.00001); and negatively correlated with hemoglobin (*p* = 0.008) and lymphocytes (*p* = 0.03). Elevated serum IL-10 levels were associated with an increased risk of anti-SS-A/Ro antibody positivity (*p* = 0.03). Elevated serum IL-6 levels were associated with an increased risk of heart (*p* = 0.007) and lung (*p* = 0.04) involvement. Conclusions: In SLE patients, increased serum IL-10 levels were associated with increased disease activity and risk of anti–SS-A/Ro antibody positivity.

## 1. Introduction

Systemic lupus erythematosus (SLE) is a chronic autoimmune disease [1,2,3]. The etiology of SLE is still not completely understood. The development of the disease is influenced by many factors, such as genetic, environmental, hormonal, and abnormal immune responses [1,4,5,6,7]. In the pathogenesis of SLE, abnormal immune responses of both cellular and humoral types play a significant role; there is a hyperactivation of B lymphocytes, dysfunction of T lymphocytes, the ratio of CD4+ T lymphocytes to CD8+ T lymphocytes is disturbed, and the activities of cytotoxic lymphocytes and natural killer (NK) cells are reduced [8]. Immune imbalance is responsible for the production of a broad spectrum of autoantibodies by B lymphocytes [9,10]. Studies published in the past 10 years have confirmed the key role of cytokines released by Th1 lymphocytes, Th2 lymphocytes, and Th17 lymphocytes in the pathogenesis of SLE [11]. 

Interleukin 10 (IL-10) is produced by Th2 lymphocytes and Treg lymphocytes, B lymphocytes, monocytes, myeloid dendritic cells, and keratinocytes [11,12]. Interleukin 10 plays a key regulatory role in inflammatory processes and cell-type responses [1,11,13]. Interleukin 10 decreases the antigen presentation capacity of antigen presenting cells (APCs), decreases the activation of Th1 lymphocytes and the pro-inflammatory cytokines produced by Th1 lymphocytes, which include interleukin 1 alpha (IL-1α), IL-6, interleukin 8 (IL-8), interleukin 12 (IL-12), TNF-α, granulocyte colony-stimulating factor (GSCF), IL-6 granulocyte colony-stimulating factor (G-CSF), and granulocyte–macrophage colony-stimulating factor (GM-CSF) [8,14]. Higher levels of IL-10 have been found in SLE patients than in healthy subjects [4,5,15,16,17,18]. IL-10 plays a pathogenic role in SLE [19].

The determination and evaluation of antinuclear antibodies (ANA) titers is the primary screening test for the diagnosis of autoimmune diseases. The presence of ANA is found in more than 95% of SLE patients [20]. The characteristic picture of ANA for SLE is a homogeneous or speckled/homogeneous type of luminescence. The isolation of soluble nuclear antigens (ENA) has allowed their differentiation and quantitative evaluation in the serum of SLE patients [21,22].

Anti-Ro (anti-SS-A/Ro) antibodies directed against a small-molecule ribonucleoprotein containing RNA and proteins of 52 kDa and 60 kDa are found in many disease entities: SLE, cutaneous lupus, Sjögren’s syndrome (SS), scleroderma, and polymyositis. The positivity of the antibody anti-SS-A/Ro is associated with clinical SLE manifestations, such as leukopenia, subacute cutaneous lupus, neonatal lupus, and congenital heart block [21,22,23].

The aim of this study was to evaluate the association between serum IL-10 levels in relation to selected autoantibodies, disease activity, and organ involvement in SLE patients. We decided to focus on IL-10 and IL-6 because of the suggested opposite mechanism of action—anti-inflammatory for IL-10 and pro-inflammatory for IL-6.

## 2. Materials and Methods

### 2.1. Patients and Controls

We studied 200 Caucasian patients with confirmed diagnoses of SLE and recorded data concerning their age, sex, disease duration, organ involvement, disease activity, and treatment. The control group consisted of 50 healthy individuals (44 females, 6 males) age- and sex-matched with the study group. The ethics committee of the Pomeranian Medical University in Szczecin approved this study (KB-0012/11/13), and all participants provided informed consent.

The diagnosis of SLE was established according to the American College of Rheumatology (ACR) criteria of 1982 (modified in 1997) and the classifications developed by the Systemic Lupus International Collaborating Clinics (SLICC) of 2012 [24]. The disease activity of SLE was assessed according to the Systemic Lupus Erythematosus Disease Activity Index (SLEDAI) scale in a modified version: SLEDAI-2000 (SLEDAI-2K) [25]. 

### 2.2. Laboratory and Serological Diagnostics

For the estimation of IL-6 and IL-10 levels, sera were stored at −80 °C until analysis using a sensitive sandwich enzyme-linked immunosorbent assay (ELISA) method using Human IL-10 Immunoassay Quantikine^®®^ ELISA kit, Human IL-6 Immunoassay Quantikine^®®^ ELISA kit from R&D Systems, United States.

IgG antinuclear antibodies (ANA) were assessed in a HEp-2 cell line contaminated by CVCL-0030 cervical adenocarcinoma human HeLa using indirect immunofluorescence assay (IIFA). Monospecific tests were also performed using the ELISA method to detect anti-double-stranded DNA (anti-dsDNA), anti-Sm, anti-SS-A/Ro, anti-SS-B/La, anti-nucleosome (anti-NuA), anti-ribosomal P protein, anti-histone, and anti-U1-RNP antibodies (EUROIMMUN AG Medizinische Labordiagnostika tests, Lűbeck, Germany). The reference values of ANA were established as absent when the titer was <1:160 and present when the titer was ≥1:160. The titers were divided into three groups: low titers from 1:160 to 1:320, medium titers from 1:640 to 1:1280, and high titers > 1:1280.

The profiles of anti-phospholipid antibodies (aPL), including anticardiolipin (aCL) and anti-beta 2 glycoprotein I (aβ2-GPI), were determined using the ELISA method (EUROIMMUN AG Medizinische Labordiagnstika tests, Lűbeck, Germany). The lupus anticoagulant (LA) was tested using coagulological methods according to criteria of the International Society of Thrombosis and Haemostasis [26].

Additionally, blood was taken for the assessment of leukocyte, lymphocyte, and platelet numbers, hemoglobin (Coulter counter), ESR (Westergren method), C-reactive protein (CRP) (turbidimetric nephelometry), fibrinogen (Clauss method), and complement factors C3 and C4 (nephelometry) levels. During the blood sample collection, all patients were on pharmacological treatments.

### 2.3. Statistical Analysis

In order to determine the data distributions, the Kolmogorov–Smirnov test was evaluated. Data are presented as means (SD) and medians (Q1, Q3). To calculate the correlation between serum IL-10, serum IL-6, and laboratory markers of disease activity we used the rank Spearman test, the R values of correlations were determined, and corresponding *p* values < 0.05 were considered significant. The groups were compared using a Student’s t-test, Mann–Whitney U test, and Kruskal–Wallis test. In order to evaluate the parameters a Pearson’s chi-squared test (χ^2^), logistic regression analysis, and stepwise analysis were performed. The *p* values < 0.05 were considered statistically significant. All statistical data were analyzed using STATA 11: license number 30110532736 (StatSoft Inc, Tulsa, OK, USA).

## 3. Results

The baseline clinical characteristics of all the patients and controls are shown in Table 1. We studied 200 patients (181 females, 19 males) with a confirmed diagnosis of SLE. There was no significant difference in gender and age between the study group and the control group (*p* > 0.05) (Table 1). 

The duration of disease was significantly longer in females than in males (*p* = 0.008) (Table 1). Analysis of SLE disease activity as measured by the SLEDAI scale showed a low activity in 66.05% of patients, moderate activity in 26.54% of patients, and high activity in 7.41% of patients (Table 1). 

The baseline laboratory characteristics of all the patients and controls are shown in Table 2. Median serum concentrations of IL-10 (*p* < 0.00001) and IL-6 (*p* < 0.00001) were higher in the study group than in the control group (Table 2).

Analysis of serological tests in the study group revealed the presence of ANA in 198 (99.0%) SLE patients. Antinuclear antibodies in high titers were found in 99 (49.5%) patients. Medium ANA titers were found in 57 (28.5%) patients. Low titers of ANA were found in 42 (21.0%) patients. Analysis of the type of luminescence of ANA antinuclear antibodies in SLE patients showed the presence of the following luminescence patterns: homogeneous (39.0%), speckled (31.0%), speckled/homogeneous (26.0%), and peripheral (3.0%) (Table 2).

In SLE patients, there was no significant correlation between serum IL-10 levels and patients’ age or disease duration (all *p* > 0.05). There was a positive correlation between serum levels of IL-10 and IL-6 (*p* < 0.00001), CRP (*p* < 0.00001), fibrinogen (*p* = 0.003), and ESR (*p* < 0.00001) levels. There was a negative correlation between serum levels of IL-10 and hemoglobin (*p* = 0.0004), and lymphocyte (*p* = 0.01) levels (Table 3). 

There was a positive correlation between serum levels of IL-6 and CRP (*p* < 0.00001), fibrinogen (*p* = 0.001), and ESR (*p* < 0.00001) levels. There was a negative correlation between serum levels of IL-6 and hemoglobin (*p* = 0.008), as well as lymphocyte (*p* = 0.03) levels (Table 3).

The cutoff values for IL-10 and IL-6 were determined based on the ROC (received operating characteristic) curves as 5.11 pg/mL and 1.53 pg/mL, respectively. The area under the ROC curves was 0.798 for IL-10 and 0.757 for IL-6. The areas under the ROC curves plotted for each studied parameter were determined to assess their clinical utility.

Elevated serum IL-10 levels > 5.11 pg/mL were found to be associated with an increased risk of anti-SS-A/Ro antibodies in SLE patients (*p* = 0.03). There was no relationship between elevated serum IL-10 levels and the presence of other antibodies in SLE patients (all *p* > 0.05) (Table 4).

In a multivariate logistic analysis model with backward stepwise analysis, there was no association between serum IL-10 levels and the occurrence of organ involvement in SLE patients (all *p* > 0.05). Nevertheless, elevated serum IL-6 levels > 1.53 pg/mL were associated with an increased risk of heart (*p* = 0.007) and lungs (*p* = 0.04) involvement (Table 5).

No differences were found in SLE groups of various treatment regiments in terms of serum IL-10 and IL-6 levels (all *p* > 0.05).

## 4. Discussion

Systemic lupus erythematosus is an autoimmune disease that entails immune system dysfunction, involving the production of antibodies and formation of immune complexes. We conducted a study of SLE patients, for whom serum levels of IL-10 and selected antibodies were evaluated in association with disease activity and organ involvement. 

Interleukin 10 is an anti-inflammatory cytokine that affects the synthesis of autoantibodies by B lymphocytes, and its involvement in the pathogenesis of SLE is considered. The increased production of IL-10 in SLE patients leads to the activation of B lymphocytes and to the production of autoantibodies by B lymphocytes, and immune complexes formation. Thus, it can be concluded that elevated IL-10 levels may be an important factor that plays a role in the pathogenesis of SLE by enhancing and perpetuating the inflammatory process [17].

In our study, we observed higher serum IL-10 levels in SLE patients compared to controls, which is consistent with previous findings by other authors [4,8,16,17,18,27]. 

We did not observe a correlation between serum IL-10 levels and patient age or disease duration, which is consistent with the results of previous studies [17].

Interleukin 10 in SLE patients has pro-inflammatory properties. Unlike in other studies, we did not find a correlation between serum IL-10 levels and clinical disease activity as measured by SLEDAI [4,8,15,16,17,18]. This may be due to the fact that most of the subjects had a low disease activity, as measured by SLEDAI. On the other hand, as with other researchers, we revealed an association between serum IL-10 levels and laboratory markers of disease activity, such as CRP, fibrinogen, IL-6, ESR, and lymphocytes [16,17,18]. In our study, we found a negative correlation between serum IL-10 levels and lymphocyte counts. These results are consistent with the study by Dima A et al. [28]. The increased production of IL-10 in SLE patients leads to the activation of lymphocyte apoptosis, which affects the formation of autoantibodies and contributes to disease activity and progression.

The association between serum IL-10 levels and the positivity of anti-dsDNA antibodies were observed. Some researchers found the positivity of anti-dsDNA antibodies in SLE patients at higher concentrations of serum IL-10, while others showed a positive correlation between the serum levels of IL-10 and the titers of anti-dsDNA antibodies [29,30]. However, our study did not show a correlation between serum IL-10 levels and the positivity of anti-dsDNA antibodies, which was likely related to the selection of the study group, as suggested by other researchers [17]. Interestingly, in our study, we found that elevated serum IL-10 levels (>5.11 pg/mL) were associated with an increased risk of anti-SS-A/Ro antibody positivity in SLE patients. Anti-SS-A/Ro antibodies are a well-known risk factor for neonatal lupus. The most severe complication of neonatal lupus is complete heart block [31,32]. In a study by Diaz T. et al. [32], it was found that women with SLE with a high titer of anti-SS-A/Ro antibodies had a 1–5% risk of having their first child with congenital atrioventricular block III°. When a subsequent child is born, this risk would increase to 18% [32]. In the available literature, we did not find any studies presenting an association between serum IL-10 levels and the positivity of anti-SS-A/Ro antibodies in SLE patients. Our results indicate the validity of further studies on women with SLE that are necessary to evaluate the relationship between serum IL-10 levels and the positivity of anti-SS-A/Ro antibodies. Perhaps the determination of serum IL-10 in conjunction with the determination of anti-SS-A/Ro antibodies could be a useful marker for assessing the risk of neonatal lupus. 

The increased production of IL-10 in SLE patients leads to the activation of lymphocyte apoptosis, the formation of autoantibodies, and disease progression [19]. However, no association was observed between serum IL-10 levels and the risk of organ involvement in SLE patients.

Interleukin 6 is a pro-inflammatory cytokine that is involved in the activation of the immune system and its response [33]. This cytokine plays a key role in hematopoiesis, inflammation regulation, and acute phase proteins production [32]. In our study, in SLE patients, higher levels of IL-6 compared to controls were observed. This is consistent with previous findings and confirms the role of IL-6 in SLE pathogenesis [4,27,34]. We showed a positive correlation between serum IL-6 levels and the age of SLE patients, while there was no correlation between serum IL-6 levels and disease duration. Studies conducted by other authors have revealed that serum IL-6 levels increase with the aging of the immune system [35,36,37,38,39]. This process is associated with progressive and irreversible changes in immune response, including changes in the regulation and production of pro-inflammatory and anti-inflammatory cytokines. With age, there is a decrease in immune responses, an increase in the occurrence of inflammation and oxidative stress, and an increase in the production and release of autoantibodies [39].

The association between serum IL-6 levels and disease activity in SLE patients is not clear. Similar to other investigators, in our study, we did not find a significant correlation between serum IL-6 levels and disease activity as measured by SLEDAI [31,38]. Nevertheless, some other studies have shown a positive correlation between serum IL-6 levels and SLEDAI [4,34,40,41]. 

During the acute phase response, IL-6 (produced by immune cells) is responsible for the production of acute phase proteins by liver cells, such as serum amyloid A (SAA), α1-antitrypsin, fibrinogen, and CRP [39,42]. This was confirmed by our study showing a positive correlation between serum IL-6 levels and inflammatory markers such as CRP, fibrinogen, and ESR, and this is consistent with other studies [16,39,42]. Nevertheless, in other studies, no correlation between serum IL-6 levels and CRP or ESR was found [43].

In our study, we found a positive correlation between serum IL-6 levels and IL-10 levels, which is in agreement with the study by Yin Z. and associates [16].

In a study by Abdel Galil S.M. et al. [41] on patients with lupus nephropathy, the group with the active disease showed a positive correlation between serum IL-6 levels and anti-dsDNA antibody titers, in contrast to patients with inactive SLE. Another study showed that SLE patients with positivity of anti-dsDNA antibodies had higher serum IL-6 concentrations compared to SLE patients without anti-dsDNA antibodies [44]. Unlike the described reports, in our study, we did not find a relationship between serum IL-6 levels and the positivity of anti-dsDNA antibodies [39,44]. This is likely because 54.3% of our study group had a negative result for anti-dsDNA antibodies.

In SLE patients, we demonstrated an association between elevated serum IL-6 levels and the occurrence of anemia. Analysis of the data showed a negative correlation between serum IL-6 levels and hemoglobin concentration. The effect of elevated IL-6 levels on the suppression of erythropoiesis in SLE patients was confirmed by other studies, which showed higher serum IL-6 levels in SLE patients with anemia compared to SLE patients without anemia [40,41]. There was also a negative correlation between serum levels of IL-6 and hemoglobin in SLE patients with renal involvement [41]. Thus, it can be concluded that serum IL-6 levels may be an important factor playing a role in the pathogenesis of anemia in SLE patients.

The involvement of multiple organs, such as in skin lesions, arthritis, and changes in the cardiovascular system, kidneys, lungs, and nervous system, can be observed in the course of SLE. We do not have specific markers with which to predict the exacerbation or remission of organ lesions in the course of SLE. A higher occurrence of cardiovascular disorders in SLE patients compared to the healthy population of the same sex and age was observed in up to 50% of patients [45,46]. In our research, cardiovascular disease was found in 40.4% of SLE patients. Cardiovascular involvement in SLE patients can entail the pericardium, endocardium, myocardium, valvular apparatus of the heart and coronary vessels [47]. Our research has shown that, in SLE patients, elevated serum IL-6 levels (>1.53 pg/mL) are associated with an increased risk of constitutional symptoms (fever), lung disease, and cardiovascular changes. Data show that, in SLE patients, elevated serum IL-6 levels were associated with constitutional symptoms, especially fever [48,49]. Studies by other authors also found elevated serum IL-6 levels in SLE patients with heart failure, coronary artery disease, and ischemic heart disease [50,51]. Chronic inflammation induced by IL-6 can lead to vascular endothelial dysfunction, induce myocardial changes and increase mortality in the course of SLE [45,52].

In SLE patients, the influence of various treatment regimes in terms of serum IL-10 and IL-6 levels was not observed. 

We are aware of the limitations of our study. In our study, we have presented the preliminary, statistically significant results concerning association between serum IL-10 concentration and anti-SS-A/Ro antibody positivity in SLE, which we believe point the way towards further studies involving larger groups of patient. There are no data in the available literature on such an association. In our opinion, the results obtained may be useful in assessing the risk of neonatal lupus. 

In addition, the disease duration of most of the cases studied was between 5 and 10 years, so we were unable to determine the relationship between serum IL-10 levels and the risk of the anti-SS-A/Ro antibody positivity in patients with early-stage of SLE. Therefore, the investigation of the above relationships in connection with the monitoring the differences of immune cell populations in SLE patients at different stages of the disease will be considered the subject of our future studies. 

## 5. Conclusions

In patients with SLE, increased serum IL-10 levels are associated with increased disease activity and increased risk of anti-SS-A/Ro antibody positivity, and elevated serum IL-6 levels are associated with an increased risk of heart and lung involvement.

We suppose that the determination of serum IL-10 in combination with the determination of anti-SS-A/Ro antibody could be considered as a useful biomarker for assessing the risk of neonatal lupus.

Considering the limitations of our study, the investigation of the above relationships in SLE patients at different stages of the disease will be considered in our future studies.

## Figures and Tables

**Table 1 biomolecules-13-00974-t001:** Clinical characteristics of systemic lupus erythematosus patients and healthy controls.

Assessed Parameters			Study Group n = 200 Mean ± SD Number (%)	Control Group n = 50 Mean ± SD Number (%)	*p* Value
Sex		F	181 (90.5)	44 (88.0)	0.6
		M	19 (9.5)	6 (12.0)	
Age (years)		Study group	46.97 ± 13.73	42.72 ± 12.48	-
		F	46.00 ± 12.86	42.00 ± 17.24	-
		M	52.00 ± 14.52	45.00 ± 10.34	-
		≤40 years	67 (33.5)	19 (38.0)	0.5
		>40 years	133 (66.5)	31 (62.0)	
Disease duration (years)		All group	10.40 ± 9.10	-	-
		F	11.00 ± 9.32	-	
		M	5.11 ± 3.53	-	0.008
SLEDAI scale		≤11	107 (66.05)	-	-
		>11 ≤ 20	43 (26.54)	-	-
		>20	12 (7.41)	-	-
Clinical presentation				-	-
Malar rash			115 (57.50)	-	-
Discoid rash			12 (6.00)	-	-
Photosensitivity			145 (72.50)	-	-
Hair loss			75 (37.50)	-	-
Oral ulcers			44 (22.00)	-	-
Arthritis			155 (77.50)	-	-
Serositis			80 (40.00)	-	-
Renal lupus			43 (21.50)	-	-
Class of lupus nephropathy on histopathological examination	• II class		11 (5.50)	-	-
	• III class		8 (4.00)	-	-
	• IV class		5 (2.50)	-	-
	• V class		5 (2.50)	-	-
Neurologic involvement			68 (34.34)	-	-
Hematologic involvement			139 (69.50)	-	-
Treatment	Cs		162 (81)	-	-
	Antimalarials		154 (77)	-	-
	Azathioprine		30 (15)	-	-
	Cyclosporin A		4 (2)	-	-
	Cyclophosphamide		43 (21.5)	-	-
	Methotrexate		7 (3.5)	-	-
	MMF		10 (5)	-	-
	Immunoglobulins		12 (6)	-	-
	Epratuzumab		2 (1)	-	-

n—number, F—female M—men, SLEDAI—Systemic Lupus Erythematosus Activity Index, Cs—corticosteroids, MMF—mycophenolate mofetil.

**Table 2 biomolecules-13-00974-t002:** Laboratory characteristics of systemic lupus erythematosus patients and healthy controls.

Examined Antibodies	Study Group n = 200 Mean ± SD Median (Q1, Q3) Number (%)	Control Group n = 50 Mean ± SD Median (Q1, Q3) Number (%)	*p* Value
IL-10 (pg/mL)	7.35 (4.90, 11.18)	3.36 (2.34, 5.08)	<0.00001
IL-6 (pg/mL)	2.50 (0.89, 5.40)	0.84 (0.30, 1.26)	<0.00001
ESR (mm/h)	16.00 (8.00, 30.00)	6.00 (4.00, 10.00)	<0.0001
CRP (mg/l)	1.89 (1.00, 5.83)	-	-
WBC (10^3^/µL)	5.71 (4.36, 7.46)	5.97 (4.73, 6.65)	0.7
Lymphocytes (10^3^/µL)	1.36 (0.99, 1.82)	1.81 (1.57, 2.17)	<0.0001
HGB (g/dl)	12.67 ± 1.70	13.85 ± 1.08	<0.0001
PLT (10^3^/µL)	227.4 ± 78.2	251.2 ± 55.6	0.04
Fibrinogen (mg/dl)	349.5 ± 108.4	280.0 ± 65.5	0.0001
Complement factor C3 (mg/dl)	97.45 ± 25.2	-	-
Complement factor C4 (mg/dl)	16.86 ± 7.52	-	-
Proteinuria	38 (19.30)	-	-
Erythrocyte urinary casts	4 (2.00)	-	-
Hemolytic anemia	10 (8.50)	-	-
Deficiency anemia	82 (43.40)	-	-
Leucopenia	75 (37.69)	-	-
Lymphopenia	87 (43.50)	-	
Thrombocytopenia	42 (21.11)	-	-
Positive direct Coombs’ test	25 (27.17)	-	-
False positive syphilis test (VDRL)	2 (1.90)	-	-
Decreased concentration of complement factor C3	88 (45.13)	-	-
Decreased concentration of complement factor C4	56 (28.72)	-	-
ANA IgG	198 (99.00)	1 (2)	-
Anti-dsDNA IgG	86 (45.70)	-	-
Anti-NuA IgG	57 (32.90)	-	-
Anti-Sm IgG	9 (5.10)	-	-
Anti-SS-A/Ro IgG	68 (39.10)	-	-
Anti-SS-B/La IgG	25(14.90)	-	-
Anti-ARPA IgG	6 (3.50)	-	-
Anti-histones IgG	24(14.00)	-	-
Anti-U1-snRNP IgG	22 (12.60)	-	-
Anti-CL IgG	49 (28.00)	-	-
Anti-CL IgM	66 (37.70)	-	-
Anti-ß2-GPI screen IgA, IgG, IgM	61 (35.30)	-	-

IL-10—interleukin 10, IL-6—interleukin 6, ESR—erythrocyte sedimentation rate, CRP—C-reactive protein, WBC—white blood cell, HGB—hemoglobin, PLT—blood platelets, VDRL—Venereal Diseases Research Laboratory, ANA—anti-nuclear antibodies, Anti-dsDNA—anti-double stranded DNA antibodies, Anti-NuA—anti-nucleosome antibodies, Anti-Sm—anti-Smith antibodies, Anti-SS-A/Ro—anti-Rose antibodies, Anti—ARPA—anti-ribosomal P protein antibodies, Anti-CL—anticardiolipin antibodies, Anti—B2GP-I—β2-glycoprotein I antibodies, Ig A—immunoglobulin A, Ig G—immunoglobulin G, Ig M—immunoglobulin M, “-“—a given value was not measured.

**Table 3 biomolecules-13-00974-t003:** Results of correlation analysis between serum IL-10 and serum IL-6 and laboratory markers of disease activity in the study group.

Assessed Parameter	Levels of IL-10 (pg/mL)		Levels of IL-6 (pg/mL)	
	Spearman’s Rank Correlation Coefficient, R	*p* Value	Spearman’s Rank Correlation Coefficient, R	*p* Value
Age (years)	0.06	0.4	0.24	0.0005
Disease duration (years)	−0.08	0.2	-0.04	0.5
IL-6 (pg/mL)	0.41	<0.00001	-	-
CRP (mg/l)	0.37	<0.00001	0.51	<0.00001
ESR (mm/h)	0.34	<0.00001	0.47	<0.00001
Fibrinogen (mg/dl)	0.29	0.003	0.32	0.001
SLEDAI	0.05	0.5	0.14	0.08
WBC (10^3^/µL)	0.10	0.13	0.01	0.88
Lymphocytes (10^3^/µL)	−0.17	0.01	−0.15	0.03
HGB (g/dl)	−0.25	0.0004	−0.19	0.008
PLT (10^3^/µL)	0.09	0.22	−0.04	0.58

IL-10—interleukin 10, IL-6—interleukin 6, CRP—C-reactive protein, ESR—erythrocyte sedimentation rate, SLEDAI,—Systemic Lupus Erythematosus Activity Index, WBC—white blood cell, HGB—hemoglobin, PLT—blood platelets.

**Table 4 biomolecules-13-00974-t004:** A logistic regression model for the OR of the increased serum IL-10 and IL-6 levels and autoantibodies profile in patients with systemic lupus erythematosus.

Assessed Parameter	Levels of IL-10 > 5.11 pg/mL			Levels IL-6 > 1.53 pg/mL		
	OR	95% CI	*p* Value	OR	95% CI	*p* Value
Anti-Sm IgG	0.46	0.12–1.78	0.2	0.75	0.19–2.90	0.6
Anti-SS-A/Ro IgG	2.20	1.05–4.64	0.03	1.28	0.68–2.42	0.4
Anti-Ro52 IgG	1.47	0.70–3.08	0.3	1.23	0.62–2.44	0.5
Anti-SS-B/La IgG	1.55	0.55–4.42	0.4	1.33	0.54–3.28	0.5
Anti-dsDNA IgG	1.54	0.79–2.98	0.2	1.51	0.83–2.75	0.1
Anti-NuA IgG	1.16	0.57–2.37	0.6	1.72	0.87–3.39	0.1
Anti-histones IgG	1.61	0.56–4.58	0.3	1.61	0.63–4.12	0.3
Anti-ARPA IgG	0.80	0.14–4.50	0.7	0.60	0.12–3.08	0.5
aCL IgG	1.23	0.58–2.63	0.5	1.95	0.94–4.02	0.07
aCL IgM	1.48	0.73–2.99	0.2	1.62	0.85–3.09	0.1
aß2-GPI screen IgA, IgG, IgM	0.91	0.46–1.82	0.7	1.06	0.56–2.03	0.8

OR—Odds ratio, adjusted for gender and age; 95% CI—confidence interval, Anti-Sm—anti-Smith antibodies, Anti-SS-A/Ro—anti-Rose antibodies, Anti-dsDNA—anti-double stranded DNA antibodies, Anti—ARPA—anti-ribosomal P protein antibodies, aCL—anticardiolipin antibodies, a B2GP-I—β2-glycoprotein I antibodies, Ig A—immunoglobulin A, Ig G—immunoglobulin G, Ig M—immunoglobulin M.

**Table 5 biomolecules-13-00974-t005:** A logistic regression model for the OR of the increased serum IL-10 and IL-6 levels and organ involvement in patients with systemic lupus erythematosus.

Organ Manifestations	Levels of IL-10 > 5.11 pg/mL			Levels IL-6 > 1.53 pg/mL		
	OR	95% CI	*p* Value	OR	95% CI	*p* Value
Constitutional	1.73	0.79–3.76	0.1	2.02	0.99–4.12	0.05
Mucocutaneous	0.67	0.33–1.37	0.2	1.21	0.65–2.27	0.5
Musculoskeletal	1.36	0.67–2.79	0.3	1.36	0.70–2.65	0.3
Heart	1.39	0.72–2.69	0.3	2.36	1.27–4.41	0.007
Myocardial infarction	3.36	0.41–27.22	0.2	5.73	1.46–252.38	0.01
Ischemic heart disease	6.65	0.86–51.63	0.07	9.60	1.24–74.24	0.03
Hypertension	1.94	0.93–4.06	0.07	1.73	0.90–3.33	0.1
Lung	3.57	0.80–15.96	0.09	3.57	1.01–12.63	0.04
Hematological	0.78	0.39–1.55	0.4	1.04	0.56–1.92	0.9
Vascular system	0.94	0.39–2.27	0.8	2.44	0.94–6.32	0.06
Nervous system	0.64	0.32–1.27	0.1	1.35	0.69–2.61	0.3
Kidneys	1.91	0.89–4.11	0.09	1.25	0.61–2.55	0.5

OR—Odds ratio, adjusted for gender and age, 95% CI—confidence interval, IL-10—interleukin 10, IL-6—interleukin 6.

## Data Availability

The data used to support the findings of this study are available from the corresponding author upon request.

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
