# Peer review of "Increased Serum Interleukin 10 Levels Are Associated with Increased Disease Activity and Increased Risk of Anti-SS-A/Ro Antibody Positivity in Patients with Systemic Lupus Erythematosus"

_biomolecules, 2023, doi:10.3390/biom13060974_

Round 1

Reviewer 1 Report

The study aimed at measuring correlation between the IL-10 and IL-6 serum levels and other clinical characteristics and SLE. It is a well planned and conducted medical research with the results consistent with other published studies. The majority of patients though had low-to medium indices as assessed by SLEDAI-2K, therefore it definitely affected the results and authors are aware of this limitation. 

It is not clear whether all patients were on treatment during the blood sampling or there were some not on medication. The pharmacological treatment itself may contribute to the alterations in the serological parameters. I suggest to clarify this in the Materials and Methods section and discuss it and in the main text.    

Table 2 shows clinical characteristics in patients and controls. The column with the control group lacks many such characteristics, is it because the values were very low in the controls, or because those parameters were not measured in the controls? Please, state it clearly in the Table legend.  

Minor comment:

Line 62, a typo in the word “extractable”. 

Author Response

  • It is not clear whether all patients were on treatment during the blood sampling or there were some not on medication. The pharmacological treatment itself may contribute to the alterations in the serological parameters. I suggest to clarify this in the Materials and Methods section and discuss it and in the main text.

Answer:

In Material and Methods we added:

During the blood samples collection all patients were on pharmacological treatments.

In Discussion we added:

In SLE patients the influence of various treatment regimes in terms of serum IL-10 and IL-6 levels was not observed.

  • Table 2 shows clinical characteristics in patients and controls. The column with the control group lacks many such characteristics, is it because the values were very low in the controls, or because those parameters were not measured in the controls? Please, state it clearly in the Table legend.  

Answer:

In Table 2 legend we added:

“-“ - a given value was not measured

Line 62  “extractable” change to “soluble”.

Reviewer 2 Report

The study screened the expression of IL-10 and IL-6 with other factors in the serum of SLE patients compared to normal and concluded that IL-10 level increases among SLE patients. They showed a positive relation with IL-6 and increased risk of anti-SS-A/RO antibody. 

Actually, this study didn't show any novel or interesting findings. The increased levels of IL-10 and IL-6 among SLE patients are well known and reported by tens of literature. 

I suggest authors to investigate the relation between IL-10 levels and IL-6 at different stages of the disease and monitor the differences of immune cell populations and transcriptomics. 

Furthermore, I suggest observing the IL-10 levels during the course of SLE treatment with different types of drugs and investigating the association with response level for those drugs that would be interesting to improve the novelty of this study. 

Mostly good. 

Author Response

  • Actually, this study didn't show any novel or interesting findings. The increased levels of IL-10 and IL-6 among SLE patients are well known and reported by tens of literature. 

Answer:

We agree that the data on elevated IL-10 and IL-6 levels in SLE patients are well-known. However, the novelty of our work, which we pointed out in the conclusion, was that:
In patients with SLE, increased serum IL-10 levels are associated with increased disease activity and increased risk of anti-SS-A/Ro antibody positivity, and elevated serum IL-6 levels are associated with an increased risk of heart and lung involvement.

We suppose that the determination of serum IL-10 in combination with the determination of anti-SS-A/Ro antibody could be considered as a useful biomarker for assessing the risk of neonatal lupus.

  • I suggest authors to investigate the relation between IL-10 levels and IL-6 at different stages of the disease and monitor the differences of immune cell populations and transcriptomics. 

Answer:

Investigation of suggested relation between IL-10 levels and IL-6 at different stages of the disease and monitoring the differences of immune cell populations and transcriptomics will be the subject of our further studies.

  • Furthermore, I suggest observing the IL-10 levels during the course of SLE treatment with different types of drugs and investigating the association with response level for those drugs that would be interesting to improve the novelty of this study. 

Answer:

As we mentioned in “Results” in SLE patients the influence of various treatment regimes in terms of serum IL-10 and IL-6 levels was not observed (all p > 0.05). This was probably related to the low disease activity in most studied patients. Nevertheless we plan to follow up patients of our study to assess the relation between Il-10 levels and IL-6 at different stages of the disease.

Reviewer 3 Report

In this study, the authors aimed to evaluate the association between serum IL-10 levels and autoantibodies, disease activity and organ involvement in patients with SLE. Thus, they showed that increased serum IL-10 levels were associated with increased disease activity and risk of anti–SS-A/Ro antibody positivity. Although this is an interesting study, major problems were found with the analysis method, and the study is considered to lack novelty. The following are important issues to address.

1)        The reason for focusing only on IL-10 and IL-6 among many cytokines is unclear.

2)        The rationale for setting the cut-off values for IL-10 (5.11 pg/ml) and IL-6 (1.53 pg/ml) is not provided.

3)        The present method of analysis may not be sufficient to describe the relationship between IL-10 and each factor because of the strong correlation between IL-6 and IL-10.

4)        The authors mentioned that most of the results of the present study have been reported previously, and it is not clear whether the association between IL-10 and anti-SS-A antibodies is truly significant only in the present study, given the influence of multiple comparisons in Table 4.

Minor editing of English language required.

Author Response

  • The reason for focusing only on IL-10 and IL-6 among many cytokines is unclear.

Answer:

We decided to focus on IL-10 and IL-6 because of the suggested opposite mechanism of action in the course of SLE - anti-inflammatory for  IL-10 and pro-inflammatory for IL-6.

In Introduction we added:

We decided to focus on IL-10 and IL-6 because of the suggested opposite mechanism of action - anti-inflammatory for  IL-10 and pro-inflammatory for IL-6.

  • The rationale for setting the cut-off values for IL-10 (5.11 pg/ml) and IL-6 (1.53 pg/ml) is not provided.

Answer:

The cut-off values for IL-10 and IL-6 were determined based on the ROC (Received Operating Characteristic) curves as 5,11 pg/ml and 1,53 pg/ml, respectively.  The area under the ROC curves was 0,798 for IL-10 and 0,757 for IL-6. The areas under the ROC curves plotted for each studied parameter were determined to assess their clinical utility (Figure 1, Table 1)

Figure 1. ROC curves for studied parameters IL-10 and IL6 in study group

Table 1. Area Under the Curve for studied parameters IL-10 and IL6 in study group

Area Under the Curve

Parameters

AUC

95%

CI

SE

p

IL6

0,757

0,696

0,818

0,031

0,0000

IL10

0,798

0,728

0,868

0,036

0,0000

We added in Results:

The cut-off values for IL-10 and IL-6 were determined based on the ROC (Received Operating Characteristic) curves as 5,11 pg/ml and 1,53 pg/ml, respectively.  The area under the ROC curves was 0,798 for IL-10 and 0,757 for IL-6. The areas under the ROC curves plotted for each studied parameter were determined to assess their clinical utility.

  • The present method of analysis may not be sufficient to describe the relationship between IL-10 and each factor because of the strong correlation between IL-6 and IL-10.

Answer:

We are aware of the strong correlation between IL-10 and IL-6 levels, however we believe that we were able to present the relationship between IL-10 and each factor in a way that was statistically significant.

  • The authors mentioned that most of the results of the present study have been reported previously, and it is not clear whether the association between IL-10 and anti-SS-A antibodies is truly significant only in the present study, given the influence of multiple comparisons in Table 4.

Answer:

In Discussion we added:

In our study, we have presented the preliminary, statistically significant results concerning association between serum IL-10 concentration and anti-SS-A/Ro antibody positivity in SLE, which we believe point the way towards further studies involving larger groups of patient. There are no data in the available literature on such association. In our opinion, the results obtained may be useful in assessing the risk of neonatal lupus.

Round 2

Reviewer 2 Report

(1) Please add the limitations of your study at the end of your discussion section. And mention the needed further investigations.  

(2) In all tables, please write "P value" instead of P. 

(3) In the methods section, "Statistical analysis" can you write the purpose of every analysis for example: In order to measure ............... Data distributions were evaluated using the Kolmogorov–Smirnov test. 

The R values of correlations were also determined to .........

(4) As noticed in Table 1: most of the cases in this study suffering for the disease 5 -10 Years, means that we don't know the relation between IL-10 and anti-SS-A/Ro antibody in the early diagnosis of SLE patients. And age categories also. Can you add these points in the limitation of your study to be considered in the future studies. 

Author Response

Answers to Revision 2

  • Please add the limitations of your study at the end of your discussion section. And mention the needed further investigations.

Answer:

In discussion we added:

We are aware of the limitations of our study. In our study, we have presented the preliminary, statistically significant results concerning association between serum IL-10 concentration and anti-SS-A/Ro antibody positivity in SLE, which we belive point the way towards further studies involving larger groups of patient. There are no data in the available literature on such association. In our opinion, the results obtained may be useful in assessing the risk of neonatal lupus.

In addition, the disease duration of most of the cases studied was between 5 - 10 years, so we were unable to determine the relationship between serum IL10 levels and the risk of the anti-SS-A/Ro antibody positivity in patients with early-stage of SLE. Therefore, the investigation of the above relationships in connection with the monitoring the differences of immune cell populations in SLE patients at different stages of the disease will be considered the subject of our future studies.

  • In all tables, please write "P value" instead of P. 

Answer:

In all tables, we  have written „P value” instead of P

  • In the methods section, "Statistical analysis" can you write the purpose of every analysis for example: In order to measure ............... Data distributions were evaluated using the Kolmogorov–Smirnov test. 

The R values of correlations were also determined to .........

Answer:

We have made modifications to the section:

  1. Materials and Methods

2.3. Statistical analysis

In order to determine the data distributions the Kolmogorov–Smirnov test was evaluated. Data are presented as means (SD) and medians (Q1, Q3). To calculate the correlation between serum IL-10, serum IL-6, and laboratory markers of disease activity we used the rank Spearman test, the R values of correlations were determined, and corresponding p values < 0.05 were considered significant. The groups were compared using a Student’s t-test, Mann–Whitney U test, and Kruskal–Wallis test. In order to evaluate the parameters a Pearson’s chi-squared test (χ2), logistic regression analysis, and stepwise analysis were performed. The p values < 0.05 were considered statistically significant. All statistical data were analyzed using STATA 11: license number 30110532736 (StatSoft Inc, Tulsa, Oklahoma, United States).

  • As noticed in Table 1: most of the cases in this study suffering for the disease 5 -10 Years, means that we don't know the relation between IL-10 and anti-SS-A/Ro antibody in the early diagnosis of SLE patients. And age categories also. Can you add these points in the limitation of your study to be considered in the future studies. 

Answer:

In conclusions we added:

Considering the limitation of our study, the investigation of the above relationships in SLE patients at different stages of the disease will be considered in our future studies.

Reviewer 3 Report

The authors responded appropriately to my comments.

Author Response

Answers to Revision 3

  • The authors responded appropriately to my comments.

Answer:

Thank You very much for accepting our answers.